# Functional Characterization of the Dopaminergic Psychostimulant Sydnocarb as an Allosteric Modulator of the Human Dopamine Transporter

**DOI:** 10.3390/biomedicines9060634

**Published:** 2021-06-02

**Authors:** Shaili Aggarwal, Mary Hongying Cheng, Joseph M. Salvino, Ivet Bahar, Ole Valente Mortensen

**Affiliations:** 1Department of Pharmacology and Physiology, Drexel University College of Medicine, Philadelphia, PA 19102, USA; sa3295@drexel.edu; 2Department of Computational and Systems Biology, School of Medicine, University of Pittsburgh, Pittsburgh, PA 15260, USA; hoc2@pitt.edu (M.H.C.); bahar@pitt.edu (I.B.); 3The Wistar Institute, Philadelphia, PA 19104, USA; jsalvino@wistar.org

**Keywords:** dopamine transporter, allosteric modulation, transport activity

## Abstract

The dopamine transporter (DAT) serves a critical role in controlling dopamine (DA)-mediated neurotransmission by regulating the clearance of DA from the synapse and extrasynaptic regions and thereby modulating DA action at postsynaptic DA receptors. Major drugs of abuse such as amphetamine and cocaine interact with DATs to alter their actions resulting in an enhancement in extracellular DA concentrations. We previously identified a novel allosteric site in the DAT and the related human serotonin transporter that lies outside the central orthosteric substrate- and cocaine-binding pocket. Here, we demonstrate that the dopaminergic psychostimulant sydnocarb is a ligand of this novel allosteric site. We identified the molecular determinants of the interaction between sydnocarb and DAT at the allosteric site using molecular dynamics simulations. Biochemical-substituted cysteine scanning accessibility experiments have supported the computational predictions by demonstrating the occurrence of specific interactions between sydnocarb and amino acids within the allosteric site. Functional dopamine uptake studies have further shown that sydnocarb is a noncompetitive inhibitor of DAT in accord with the involvement of a site different from the orthosteric site in binding this psychostimulant. Finally, DA uptake studies also demonstrate that sydnocarb affects the interaction of DAT with both cocaine and amphetamine. In summary, these studies further strengthen the prospect that allosteric modulation of DAT activity could have therapeutic potential.

## 1. Introduction

The dopamine transporter is a membrane-bound protein present in the presynaptic terminals of dopaminergic neurons in the central nervous system. It belongs to the solute carrier 6 (SLC6) family of transporters and controls the signal amplitude and duration of dopaminergic neurotransmission by transporting extracellular (EC) dopamine (DA) from the synapse back into the presynaptic neuronal terminals [1]. Hence, the pharmacological intervention of DAT modulates the neuronal dopaminergic activity. DAT is known to be the primary site of action for several psychostimulants and recreational drugs, including cocaine, amphetamine, and methamphetamine. The stimulatory action of these drugs of abuse is caused by their interaction with DAT causing a blockade or reversal of DA transport, thereby resulting in an increase in synaptic dopaminergic levels and neurotransmission [2].

In an effort to develop therapeutics for substance use disorders, a plethora of DAT ligands have been explored to date that can competitively or noncompetitively block the interaction of cocaine and other DAT-interacting psychostimulants. However, the successful application of such compounds have largely failed due, in part, to their own abuse liability or stimulatory effects and their off-target effects that have impeded their clinical development [3]. DAT ligands such as WIN-35428 and RTI-55, which are structurally rigid, metabolically stable, and higher-affinity cocaine analogs, display pharmacological effects similar to cocaine. Such ligands are referred to as typical DAT inhibitors and suffer from abuse liability. Over the years, several potent and selective atypical DAT inhibitors have been discovered that lack cocaine-like abuse potential [4]. For example, GBR-12909, benztropine, JHW-007, bupropion, and modafinil lack the psychostimulatory effect and abuse liability of cocaine despite their ability to inhibit DA reuptake and display “atypical” behavioral effects [5,6,7,8]. These findings have spurred mechanistic studies toward determining more details on the structural biology of DAT, the mechanism of DA transport, and how various ligands interact with DAT at the molecular level [9]. 

Structurally, DAT is a 12-transmembrane domain (TMD) protein mediating DA transport driven by the sodium gradient. The substrate translocation follows an “alternating access” mechanism where the transporter sequentially transitions through outward-facing (OF), occluded, and inward-facing (IF) states to transport substrate from EC to intracellular milieu, and the OF and IF states may further be open or closed depending on the local conformation of pairs of gating residues [9,10]. The X-ray crystal structures of the bacterial (*Aquifex aeolicus*) leucine transporter (LeuT) [11], *Drosophila melanogaster* DAT (dDAT) [12], and human serotonin transporter (SERT) [13] co-complexed with a variety of ligands [14,15,16,17,18,19] have provided detailed insight into the structural biology of this family of monoamine transporters, mainly 12 α-helical TMDs (TM1-TM12) connected with flexible intracellular and EC loops with the N- and C-termini lying in the intracellular region [20]. 

The primary site for binding the endogenous substrate and other psychostimulants/drugs is the orthosteric site S1 at the core of the translocation pathway located between TM1 and TM6. Interestingly, in addition to the S1 site, both structural and functional studies have suggested that these transporters might harbor at least one additional binding site with potential allosteric effects [4,21,22,23]. For example, co-crystal structures of LeuT with different ligands bound within the EC vestibular region, which lies in the solvent-accessible pathway connecting the EC milieu to the orthosteric S1 site, provided compelling evidence of the presence of additional binding sites other than the S1 site [15,24,25]. The discovery of human SERT crystal structure with a (S)-citalopram molecule bound in the S1 site and another (S)-citalopram bound in the EC domain further convincingly proved the existence of such secondary or allosteric sites in the mammalian transporters as well [13]. Previously reported studies from our group have provided experimental evidence of the presence of another allosteric site on SERT [26]. Furthermore, the first high-affinity S2-bound allosteric ligand of SERT was recently reported and was found to potentiate the effects of the S1 ligand in both in vitro and in vivo experiments [27]. In addition, a recent study identified a novel extracellular allosteric modulator site on human glycine transporter GlyT2 (part of SLC6A5 group) that is targeted by bioactive lipids [28]. 

Through structure/function studies, molecular dynamics (MD) simulations, and comparative genomics techniques, we identified a unique allosteric domain in DAT and SERT located outside the central translocation pathway that dictated the pharmacological differences between the human transporters and transporters from the parasite *Schistosoma mansoni* [29]. In the case of human DAT (hDAT), we used the hybrid structure-based (HSB) method to successfully identify an allosteric modulator of hDAT (KM822) that interacts with this domain in the EC vestibule [30]. We named the KM822-binding site as the A2 site. Through a series of assays, we demonstrated that KM822 displayed remarkable hDAT modulating effects by reducing the cocaine potency in inhibiting the hDAT-mediated DA reuptake, as well as cocaine-associated behaviors, in a planarian model of psychostimulant activity. We further employed biochemical assays to directly prove KM822′s interaction with site A2. Collectively, these findings strongly suggested the therapeutic relevance of specifically targeting the EC vestibular region of hDAT, which contains the allosteric site A2. 

In the current study, we have provided evidence that the dopaminergic psychostimulant sydnocarb also interacts with the allosteric site A2 of hDAT, highlighting the pharmacological relevance of engaging this region of hDAT. Sydnocarb, also known as mesocarb, is a psychomotor stimulant structurally related to D-amphetamine [31]. It was developed in Russia, where it was used to treat some neuropsychiatric disorders, including schizophrenia and depression. Unlike D-amphetamine, neither significant toxic episodes nor abuse potential have been reported with sydnocarb in animals [32,33,34]. Moreover, clinically, sydnocarb’s stimulating effects are known to produce more gradually and last longer that D-amphetamine without the euphoric effects. In fact, no behavioral or physical dependence on sydnocarb has ever been reported [35]. Similar observations are made in animals where sydnocarb causes a slow and gradual increase in extracellular DA in the rat striatum and nucleus accumbens relative to D-amphetamine. As a consequence, sydnocarb is also being regarded as a potential agent for treating cocaine abuse as a nonabused, low-toxic, psychomotor stimulant drug [36]. One of the most prominent pharmacological characteristics compared with cocaine is that sydnocarb has higher selectivity and potency for inhibiting DAT. Sydnocarb also inhibits NET and SERT, but the inhibition is much less potent compared to DAT and lacks a D-amphetamine-like ability to release DA in vitro [37]. 

In the present study, we compared the binding characteristics of sydnocarb with those of KM822 using computational modeling, and a series of biochemical assays developed previously with KM822 to reveal sydnocarb’s mechanism of DAT inhibition and effect on the potency of psychostimulants such as cocaine and amphetamine. Our results thus provided molecular information on the mechanism of DAT modulation by atypical ligands like sydnocarb and the potential role of the engagement of the allosteric site by such ligands in moduating the transport activity of DAT. Multiple runs of molecular dynamics (MD) simulations suggest sydnocarb and KM822 share a common allosteric-binding site [30], the broadly defined A2 site, indicating the therapeutic and clinical potential of this site.

## 2. Materials and Methods

### 2.1. Reagents and Drugs

Sydnocarb was a gift from Teva Pharmaceuticals (Frazer, PA, USA). Radiolabeled substrates, [^3^H]-dopamine (32.6 Ci/mmol) and [^3^H]-serotonin (23.9 Ci/mmol), were purchased from PerkinElmer (Boston, MA, USA). Cell culture media and supplements, including penicillin/streptomycin, Dulbecco’s phosphate-buffered saline (DPBS), Dulbecco’s modified Eagle’s medium (DMEM) with glucose, and scintillation fluid, were obtained from Thermo Fisher Scientific (Waltham, MA, USA). Transfection reagents TransIT-LT1 and LipoJet reagent were from Mirus Bio LLC (Madison, WI, USA) and SignaGen Laboratories (Rockville, MD, USA), respectively. Reagents for uptake assays and non-radiolabeled substrates were purchased from Sigma-Aldrich (St. Louis, MO, USA). MTSEA-biotin was purchased from Biotium, Inc. (Fremont, CA, USA). Reagents for uptake assays and non-radiolabeled substrates were purchased from Sigma-Aldrich (St. Louis, MO, USA).

### 2.2. Structural Models of WT DAT in the OF and IF States

We adopted the outward-facing open (OF*o*) conformation of hDAT stabilized in earlier simulations [38], The inward-facing open (IF*o*) conformation was constructed based on the most recent cryo-EM structure of hSERT IF*o* conformer, a full atomic version of which has been deposited in the Protein Data Bank (PDB) (PDB: 6DZZ) [18]. To this aim, we used the homology modeling protocol described earlier [38]. Briefly, for each protein, one hundred homology models were generated using MODELLER [39], and the one with the lowest (MODELLER objective function) score was selected and further refined by MD simulations.

### 2.3. Force Field Parameters for Sydnocarb and KM822

The molecular structure of sydnocarb was downloaded from the ZINC database (ZINC5751608) [40]. At pH 7, the net charge carried by sydnocarb is predicted by ZINC to be zero. The molecule structure of KM822 was taken from previous study [30]. Comparison of the chemical structures of KM822 and sydnocarb is shown in Appendix A. The force field parameters for both sydnocarb and KM822 were obtained from the CHARMM General Force Field for drug-like molecules using the web server ParamChem [41]. 

### 2.4. Docking Simulations

The binding sites and poses of sydnocarb and KM822 onto the OF*o* and IF*o* DAT conformers were assessed using the protein-ligand docking software AutoDock [42]. Docking simulations were performed following the previous protocols [38]. Briefly, Lamarckian genetic algorithm with default parameters was employed, with the maximal number of energy evaluations set to 2.5 × 10^7^. The binding energy was estimated from the weighted average of multiple binding poses at a given site observed in 100 independent runs.

### 2.5. MD Simulations

To further investigate the binding propensity of sydnocarb and KM822 to hDAT, we performed full atomic MD simulations of OF*o* DAT in the presence of sydnocarb or KM822. The initial MD simulation systems were prepared using CHARMM-GUI Membrane Builder module [43]. Briefly, the DAT OF*o* conformer was embedded into 1-palmitoyl-2-oleoyl-sn-glycero-3-phosphocholine (POPC) membrane lipids, and TIP3P waters and Na^+^ and Cl^−^ ions corresponding to 0.15-M NaCl solution were added to build a simulation box of ~110 × 110 × 118 Å^3^. One KM822 or sydnocarb was initially placed or docked near the EC vestibule. E491 was protonated and a disulfide bond was added to C180 and C189 [38]. The positions of two sodium ions and one chloride ion were adopted from wt DAT [38]. Each simulation system contained ~131,000 atoms, composed of one transporter, approximately 300 lipid molecules, and 27,000 water molecules. All simulations were performed using the package NAMD [44] (version NAMD_2.12) following the previous conventional MD simulation protocol for DAT [45]. Four (so-called Run1–Run4) and two (KM822-Run1 and KM822-Run2) independent runs of 150 ns were performed for sydnocarb and KM822, respectively.

### 2.6. Site-Directed Mutagenesis, Cell Culture, and Transfections

All single and double mutants were generated using the QuikChange (Stratagene, La Jolla, CA, USA) site-directed mutagenesis kit using human WT DAT and C306A-DAT as the background, respectively. The mutations were verified by sequencing (Genewiz, South Plainfield, NJ, USA). COS-7 and HEK-293 cells were maintained in DMEM (3.5-g/L glucose) supplemented with 10% FBS and 1% penicillin/streptomycin at 37°C with 5% CO_2_. For transient transfections, COS-7 cells were transfected using the TransIT-LT1 transfection reagent (Mirus Bio LLC, Madison, WI, USA), and HEK-293 cells were transfected using the Lipojet transfection reagent (SignaGen Laboratories, Rockville, MD, USA). 

### 2.7. Transport Kinetic Assays Using COS-7 Cells

Transport assays were performed as described previously [30]. Briefly, COS-7 cells were transfected and plated in 96-well plates. Uptake experiments were performed 2 days after transfection. The media was removed, and the cells were washed with PBS (137-mM NaCl, 2.7-mM KCl, 4.3-mM Na_2_HPO_4_, and 1.4-mM KH_2_PO_4_, pH 7.4) containing 0.1-mM CaCl_2_, 1-mM MgCl_2_, 5-mM RO 41-0960, and 100-mM ascorbic acid (PBSCM). Following washing, a constant concentration of radiolabeled substrate (50-nM [^3^H]-DA) and increasing concentrations of unlabeled substrate (0.05–100-μM DA) were added, and the uptake was allowed to continue for 10 min at room temperature. The uptake was terminated by washing twice with PBS-CM. The cells were solubilized in scintillation cocktail and counted on a microplate scintillation and luminescence counter (PerkinElmer, Waltham, MA, USA). Background was obtained from non-transfected cells and subtracted. Data were fitted to a Michaelis−Menten equation using nonlinear regression to obtain K_m_ and V_max_. 

### 2.8. Transport Inhibition Assays Using COS-7 Cells

Transiently transfected COS-7 cells (expressing hDAT, hNET, or hSERT) were plated in 96-well plates. Uptake experiments were performed 2 days later. The media was removed, and the cells were washed with PBS-CM. Following washing, the cells were incubated for 10 min with various concentrations of sydnocarb, and the uptake was initiated by adding [^3^H]-dopamine to a final concentration of 25 nM. The uptake was allowed to continue for 10 min at room temperature and was terminated by washing twice with PBS-CM. The cells were solubilized in a scintillation cocktail and counted on a microplate scintillation and luminescence counter (PerkinElmer, Waltham, MA, USA). Data were fitted to a Hill equation by a nonlinear regression analysis to obtain the IC_50_ values. 

### 2.9. Biotinylation

Biotinylation was performed as described previously [30]. Briefly, HEK-293 cells were transiently transfected and cultured to confluency in a 6-well plate. The cells were washed with ice-cold PBS-CM (pH 7.4) twice and incubated at 4 °C with vehicle or sydnocarb (0.5 or 2 μM) for 10 min, followed by the addition of MTSEA-biotin (125 μM, final concentration). After 10-min incubation at 4 °C, the biotinylation was quenched by aspirating the liquid and washing the cells with 1-mM DTT in PBS-CM, followed by a final wash with PBS-CM only. The cells were lysed in 600 μL of TNE lysis buffer (pH 7.4) containing protease inhibitors (100×) for 40 min at 4 °C. The cells were collected and centrifuged at 12,000× *g* for 10 min at 4 °C. The supernatant was collected, of which 30 μL was mixed with DTT (0.5 M, 5 μL) and LDS Nu-page (15 μL) for the total lysate sample, and saved at −20 °C. Four hundred and fifty microliters of the supernatant were incubated overnight at 4 °C with 50 μL of 50% slurry of high-capacity neutravidin agarose resin. The beads containing “biotinylated” proteins were washed with ice-cold TNE lysis three times, followed by a final wash with PBS-CM, and then mixed with LDL Nu-page (15 μL), DTT (0.5 M, 5 μL), and water (30 μL). Protein samples were separated by SDS-PAGE, transferred to PVDF membranes, and probed with HA antibodies against an N-terminal HA-tag in all the DAT cysteine constructs. Densities of he DAT bands were analyzed with an Odyssey imaging system and associated ImageStudio software (LI-COR, Lincoln, NE, USA).

### 2.10. Data Analysis

Data analyses were performed using the GraphPad Prism version 7.0 for Windows (GraphPad Software Corp., La Jolla, CA, USA). Specific details of statistical tests are given in each figure legend.

## 3. Results and Discussion

### 3.1. Pharmacology of Sydnocarb Reveals the High Potency and Selectivity of Sydnocarb as a DAT Noncompetitive Inhibitor

To characterize the pharmacology and mechanism of action of sydnocarb, we performed dose-response and DA uptake saturation assays. Previous work suggested that sydnocarb is a potent and selective DAT inhibitor when compared to the other two MATs i.e., NET and SERT [37]. Consistent with those observations, our dose-response assays against hDAT, hNET, and hSERT showed that sydnocarb was 70- and 1000-fold more potent towards hDAT than towards hNET and hSERT, respectively (Figure 1A). The potency (IC_50_ values) of sydnocarb against hDAT was 0.49 ± 0.14 µM, whereas for hNET and hSERT, it was found to be 34.9 ±14.08 µM and 494.9 ±17.00 µM, respectively. The dopamine uptake kinetic results showed that both 0.5 µM and 2.0 µM sydnocarb, in a dose-dependent manner, significantly reduced the V_max_ of the DAT-mediated [^3^H]-dopamine transport compared to the vehicle, whereas the difference in the K_M_ values was not statistically significant (Figure 1B). Such uptake kinetics are indicative of an allosteric or a noncompetitive mechanism of inhibition and suggest that sydnocarb binds a site different from the orthosteric DA substrate-binding site. 

### 3.2. Docking and MD Simulations Reveal Allosteric Binding Sites for Sydnocarb 

To assess the binding propensity of sydnocarb, we first performed docking simulations onto hDAT in the outward-facing open (OF*o*) and inward-facing open (IF*o*) states using AutoDock [42]. Notably, in the case of the OF*o* conformer, the most favorable binding sites for sydnocarb were exclusively within the EC vestibule. In contrast, no high-affinity binding site was observed in the EC vestibule in the case of the IFo conformer. Therefore, we concluded that the exogenous drug sydnocarb would predominantly bind to the EC vestibule of OF*o* DAT. We have previously reported similar results with KM822, where it was also shown that KM822 stabilizes the OF*o* DAT conformation and does not bind to the IF*o* conformation of hDAT [30]. 

Next, we focused on the binding mechanism of sydnocarb to OF*o* DAT and on the changes induced on DAT structure and dynamics. To this aim, we carried out MD simulations of hDAT OF*o* in the presence of sydnocarb and compared it to our earlier results [20,38,45] on the dynamics of dopamine- or cocaine-bound hDAT and repeated the simulations and comparative analysis with KM822. A typical setup of the MD simulations with an explicit membrane and solvent is shown in Figure 2 and Appendix A for sydnocarb and KM822, respectively. In all simulation runs, the root mean square deviation (RMSD) from the original hDAT structure attained 2.5 ± 0.4 Å around 50–80 ns and remained almost flat during the rest of simulations, indicating that the ligand-bound DAT was structurally stable. The time evolution of ligand diffusion, as well as contacts, between hDAT and sydnocarb are presented in Figure 3 for Run1 and Run2 and Appendix A for Run3 and Run4. Therein, the ordinate lists the residues that come into contact with sydnocarb as it binds and settles into a stable position as a function of time (abscissa). Notably, three independent MD runs (Run2-4) resolved a similar binding site for sydnocarb that broadly agreed with the allosteric site A2 discovered previously [30]. By contrast, the sydnocarb-binding site sampled by MD simulation Run1 differed from the A2 site (see Figure 4A). For elaboration purposes, we called this the Run1-Site (see Figure 3D) to be distinguished from the main A2 site [30]. The Run1-Site was located deeper within the EC vestibule and involved residues from the TM11 and TM12 with the A2 site situated more towards the EC entrance.

To compare and contrast the sydnocarb-binding results with those of KM822, we repeated our computations with KM822 using the same protocols. KM822 also displayed multiple transiently stabilized binding poses (Appendix A), eventually stabilizing at a common binding site in both runs (Figure 4B), which coincided with the A2 site also shared by sydnocarb (Figure 4C). Interestingly, the Run1-Site was also transitionally sampled by KM822, yet KM822 proceeded to the A2 site. We anticipated that the bulkier size of KM822 made it less accessible to the Run1-Site, which was deeper within the EC vestibule compared to the A2 site. Furthermore, sydnocarb was stabilized to the Run1-Site by the π-stacking interaction with F320, together with strong hydrophobic interactions with F486, W562, and M569 (Figure 3D). By contrast, KM822 had a comparatively weaker hydrophobic interaction with W562 and A565 (Appendix A). 

### 3.3. EC Gate Closure Leading to the Transition to an Occluded state Does Not Take Place in the Presence of Sydnocarb

Our previous MD simulations of wild-type hDAT, starting from the OF*o* conformer, showed that the binding of DA and ions triggered the closure of the EC gates (R85-D476 and Y156-F320) within 60 ± 30 ns to allosterically drive the transition of the transporter to the OF closed (OF*c*) state, followed by a state occluded to both the EC and IC environments [20,38,45]. In contrast, in the case of sydnocarb, the two EC gates R85-D476 and Y156-F320 remained open in the majority of the simulations (see Figure 3 and Appendix A). These findings further suggest that the binding of sydnocarb arrests DAT in the OF open state. Similar observations were made in the docking simulations of KM822 onto the OF*o* hDAT conformation using the GOLD docking software, while the docking of KM822 onto the IFo hDAT conformation resulted in sterically unfavorable interactions [30]. In order to better visualize the multiple binding sites and poses that we observed for both KM822 and sydnocarb in our docking and MD simulations, we next examined the residues contributing to the binding of KM822 or sydnocarb within each site (Figure 4). Interestingly, the two sites broadly agree with the multiple allosteric (S2)-binding sites reported in the literature [10]. Both the Run1-Site and the A2 site share contacts with the crystal structure-resolved allosteric-binding site of (*S*)-citalopram bound to hSERT (PDB: 5I73) [13] and agreed with the sites observed for the same drug, (*S*)-citalopram, in the simulations and biochemical experiments [46]. 

Our MD simulations from KM822-Run1 showed that KM822 folds within the A2 site with the triazinoindole ring system pointing towards the EC side, making hydrophobic interactions with F217, I469, and Y548 (Figure 4C cyan and Appendix A). The arylsulfonamide-phenylacetamide moiety faced the residues W84, F155, I159, D385, and P387. Interestingly, D476 side chain made H-bond interactions simultaneously with the triazinoindole –NH, as well as the –NH of the phenylacetamide moiety. In our previously reported docking studies of KM822 using GOLD, KM822 stretched through the EC vestibule with the triazinoindole ring embedded deep inside the A2 pocket and the arylsulfonamide-phenylacetamide region facing the EC region. Our KM822-Run2 simulations in the current study resulted in a similar elongated-binding pose (Figure 4B mauve and Appendix A) within the A2 site in which the triazinoindole moiety of KM822 was surrounded by F217, P387, and Y548, and the arylsulfonamide phenylacetamide was bound deeper within the pocket towards residue F320.

When comparing the sydnocarb-binding pose within A2 (Figure 4C) with that of the KM822 pose (from KM822-Run1, Appendix A), we observed that the mesoionic moiety of sydnocarb overlapped with the arylsulfonamide-phenylacetamide region of KM822 and was surrounded by residues W84, R85, Y156, F320, and D476. The N-phenylcarbamoyl portion of sydnocarb extended towards the EC space and overlapped with the triazinoindole moiety surrounded by hydrophobic residues F217 and P387. Alternatively, the Run1-Site of sydnocarb differed significantly from the A2 location, lying deep within the EC region closer to the orthosteric region S1 (Figure 3D and Figure 4A, light orange) and was surrounded by residues R85, F320, S539, I540, W562 and M569. 

Overall, this detailed examination showed that the allosteric sites A2 and Run1-site were distinct from the substrate-binding site S1 (green surface or sticks in Appendix A) identified from the cocaine-binding simulations [38]. First, while D79 played a significant role in coordinating the binding of cocaine [38], D79 had an almost negligible binding probability to KM822 or sydnocarb in our current MD simulations. Second, whereas W84/R85 contributed to the binding of sydnocarb or KM822 with high probability (>40% simulation trajectories), W84 did not bind to cocaine. Therefore, we hypothesized that W84 (or the residues, e.g., R85, in the proximity) interacted directly with either KM822 or sydnocarb, which, thus, made W84 inaccessible to the aqueous environment; the lack of direct interaction with cocaine may render W84 accessible in a vehicle control experiment or with cocaine present.

### 3.4. Binding Site Characterization of Sydnocarb Confirms the Location of the Allosteric Site as Predicted by Docking and MD Simulations

To experimentally characterize and to confirm whether sydnocarb binds to the allosteric region of hDAT similar to that of KM822, we employed the substituted cysteine scanning mutagenesis (SCAM) method (Figure 5). We previously used this method to successfully validate the in silico-predicted binding site of KM822 to provide experimental evidence of its location and structural determinants [30]. In SCAM studies for KM822, several DAT mutants were created in which the amino acids lining the proposed allosteric pocket were systematically replaced by cysteine via site-directed mutagenesis. The only cysteine in the native DAT that was accessible from the EC environment (C306) was replaced with an alanine to create a biotinylation insensitive mutant, and this mutant (DAT-XC) was used to generate all the mutants. Then, the MTSEA-biotin reagent was used to assess the accessibility and reactivity of the thiol (-SH) group of the introduced cysteine in the absence or presence of KM822. Following biotinylation, biotinylated DAT was affinity-purified using streptavidin beads, separated, and detected using immunoblotting employing an antibody against an HA-tag that was incorporated N-terminally in all the DAT constructs. In our previous results, KM822 was consistently found to significantly protect the biotinylation of one of the mutants, W84C, which strongly suggested the proximity of KM822 with the W84 residue within the binding pocket, as seen in the current and previous computational modeling experiments. To confirm experimentally whether sydnocarb also binds to the same allosteric domain as KM822, we tested the biotinylation of the W84C mutant in the absence or presence of 1-µM sydnocarb. We observed that the biotinylation of W84C was significantly decreased by sydnocarb coincubation (Figure 5). This indicated that sydnocarb most likely binds in the vicinity of W84 similar to KM822, suggesting that sydnocarb and KM822-binding pockets overlap with each other. We used cocaine as the negative control in these experiments and observed that the labeling of W84C with MTSEA-biotin was not affected by the coincubation of 100-μM cocaine, suggesting W84 was not part of the orthosteric S1 site. This further indicated that sydnocarb and cocaine were bound at distinct binding sites within the transporter. These results provided a strong experimental validation of our MD simulations. Another possible explanation for this observation could be that the protection of the W84C residue occurred due to a long range of conformational changes caused by sydnocarb binding elsewhere. However, our modeling and experimental evidences together presented a compelling proof of direct binding. 

### 3.5. Interaction of Sydnocarb with Inward (Y335A) versus Outward (Y156F) Equilibrium Shifting DAT Mutations

Our docking experiments using AutoDock suggested that sydnocarb prefers to bind to the OF*o* conformation of hDAT, mimicking the docked KM822-hDAT conformation. In order to provide experimental evidence for these observations, we used the mutants Y156F and Y335A to determine the preferred DAT conformation for various DAT-interacting compounds [47,48,49]. Mutation Y156F removed the interaction of the Y156 hydroxyl group with D79 in the S1-binding site of hDAT, which resulted in an open-outward conformation of DAT. The binding affinities of cocaine and its analogs were unaffected by this mutation, but the potency of the atypical DAT inhibitors such as JHW-007 and S-modafinil was significantly reduced. On the other hand, the Y335A mutation shifted the DAT conformation equilibrium toward an inward-facing orientation, which markedly impaired the potency of cocaine and its analogs but caused only a slight loss in the potency of most benztropine analogs. To experimentally test whether sydnocarb preferred the outward- or the inward-facing DAT conformation, we tested its potency in a DA uptake inhibition assay in COS-7 cells transiently expressing WT, Y156F, or Y335A DAT mutants (Figure 6). We observed no significant changes in the potency of sydnocarb against WT-DAT and the outward-facing Y156F-DAT. However, the sydnocarb potency was reduced by 3.5-fold for the Y335A-DAT mutant when compared with WT-DAT. These results indicated that sydnocarb most likely prefers an outward-facing conformation more than an inward-facing conformation. We observed similar results for KM822 in our previous studies, where KM822′s potency was found to be comparable for WT-DAT and Y156F-DAT but significantly reduced for Y335A-DAT versus WT-DAT, indicating that KM822 also preferred the outward-facing DAT conformation [30]. These experimental results further supported and validated our molecular docking simulations that clearly indicated that sydnocarb and KM822 prefer the OF*o* hDAT conformation and that they do not bind to the inward-facing conformer. 

### 3.6. Sydnocarb Affects Psychostimulant Activity in In Vitro Studies

Next, we examined the effects of sydnocarb on the dose-dependent cocaine and amphetamine inhibition of hDAT-mediated DA transport in DAT transfected COS-7 cells. We observed that the presence of sydnocarb dose-dependently decreased the potency of cocaine in the dose-response DA transport inhibition assays (Figure 7). The DAT inhibition potency of cocaine was 0.177 ± 0.034 µM in the absence of sydnocarb but significantly reduced to 1.38 ± 0.24 µM in the presence of 0.5-µM sydnocarb and to 8.40 ± 0.99 µM in the presence of 2.0-µM sydnocarb. Similar results were observed with sydnocarb’s influence on the amphetamine inhibition of the DA transport. The dose response of the amphetamine inhibition of DAT-mediated DA transport significantly shifted towards the right in the presence of 0.5-µM and 2.0-µM sydnocarb. The potency values of amphetamine in the absence and presence of 0.5-µM sydnocarb were 0.166 ± 0.020 µM and 0.642 ± 0.094 µM, respectively, and 2.60 ± 1.3 µM in the presence of 2.0-µM sydnocarb. Collectively, our results showed that sydnocarb dose-dependently modulates hDAT function in entirely novel ways, such that it alters the interaction of psychostimulants cocaine and amphetamine with hDAT and, hence, could have clinical relevance in treating psychostimulant use disorders. 

## 4. Conclusions

Better options and new insights for treating psychostimulant use disorders are imperative, as the current treatments are generally known to be ineffective. The results presented in this paper provided strong empirical evidence that sydnocarb is an allosteric modulator of hDAT and that it could serve as a promising lead molecule for developing novel therapeutics against addiction. This study further supported the therapeutic relevance of engaging potential allosteric sites within the EC vestibular region of hDAT using small molecules. We used docking and MD simulations and biochemical methods to understand the molecular basis of the pharmacological potential of sydnocarb as an hDAT modulator and compared the results with that of KM822. Our previous studies showed that KM822, one of the molecules that binds to the allosteric domain, modulated the interaction of cocaine with hDAT in various in vitro and in vivo models of addiction. The mechanistic studies herein suggested that sydnocarb, like KM822, binds to the EC allosteric domain away from the orthosteric site of hDAT and closer to the EC salt bridge R85-D476, stabilizing the OF*o* conformation of hDAT. Sydnocarb also reduces the potency of hDAT to interact with the highly addictive psychostimulants cocaine and amphetamine in cell-based assays. Further works investigating the structure–activity relationship of sydnocarb will provide a clearer picture of the potential importance of developing sydnocarb as a tool for understanding the allosteric modulation of hDAT function. 

## Figures and Tables

**Figure 1 biomedicines-09-00634-f001:**
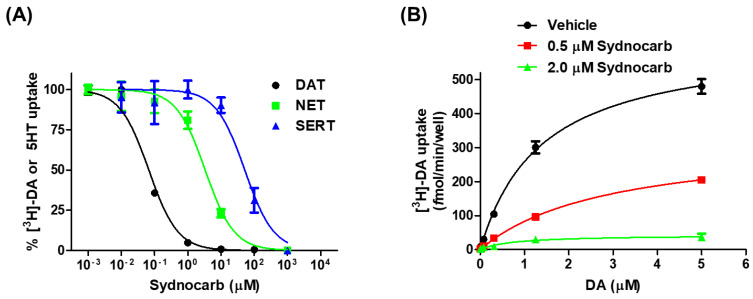
The pharmacology of sydnocarb. (**A**) Radioactive neurotransmitter transport inhibition assay of Sydnocarb against hDAT, hNET, and hSERT in transiently transfected COS-7 cells. Sydnocarb IC_50_ values for hDAT, hNET, and hSERT are 0.493 ± 0.14, 34.92 ± 14.09, and 494.87 ± 17.00 µM, respectively. The data were fitted using nonlinear regression, the figure was plotted using average of three independent experiments, and IC_50_ mean values and corresponding SEM are displayed. Results are normalized to percent of the highest response in each group. (**B**) [^3^H]-DA uptake kinetic assay of wild-type DAT transiently transfected COS-7 cells in the absence and presence of 0.5 µM and 2.0 µM Sydnocarb. Data were fitted to Michaelis-Menten equation using nonlinear regression to obtain V_max_ = 634.8 ± 84.62 for the vehicle, 455.6 ± 89.6 femtomole/min/well in the presence of 0.5 µM Sydnocarb, and 195 ± 80.7 femtomole/min/well in the presence of 2.0 µM Sydnocarb, along with the respective K_M_ values of 1.4 ± 0.21, 3.3 ± 0.78, and 1.7 ± 0.43 µM. The figure displays the average behavior from four independent experiments used to evaluate the V_max_ and K_M_. For the V_max_, the vehicle versus 0.5 µM and 2.0 µM Sydnocarb showed significant differences (*p* < 0.05) when compared using one-way ANOVA with Dunnett’s post-hoc test. No statistical significance was found in the K_M_ values.

**Figure 2 biomedicines-09-00634-f002:**
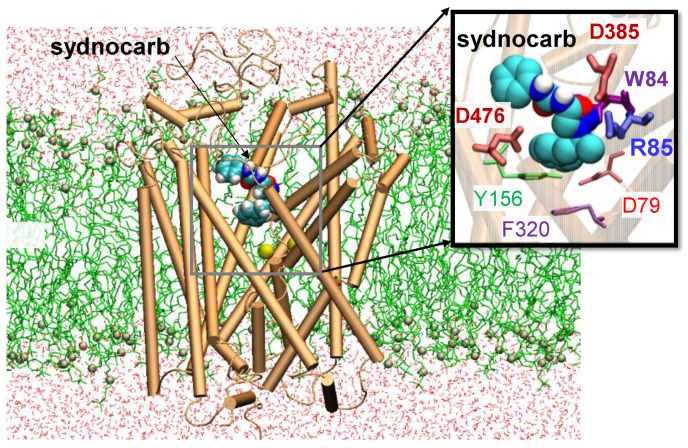
MD simulations of sydnocarb binding to the human dopamine transporter in the outward-facing open conformer. The hDAT OF*o* conformer (orange) was embedded into membrane lipids (lime licorice) and solvated by 0.15-M NaCl solution (not shown). A sydnocarb molecule (van der Waals (vdW) format) was initially docked near the EC vestibule, as predicted by AutoDock.

**Figure 3 biomedicines-09-00634-f003:**
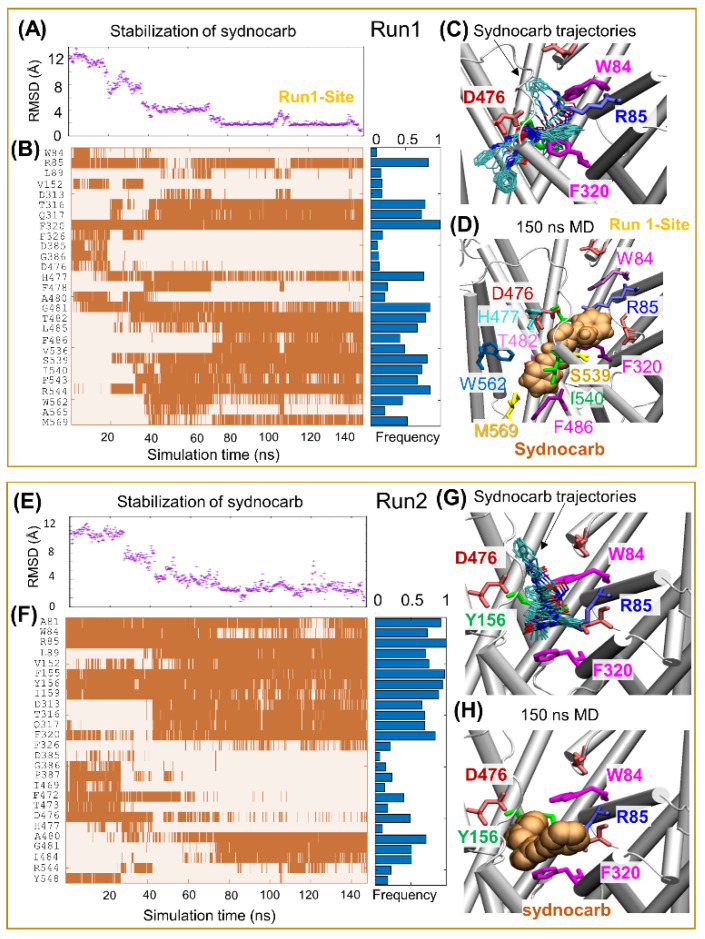
MD simulations reveal multiple binding poses of sydnocarb in hDAT. Left and right boxes display results from two independent runs: Run1 and Run2. (**A**,**E**) Sydnocarb diffusion as a function time estimated by the RMSD of sydnocarb atoms with respect to the final pose at 150 ns. (**B**,**F**) Time evolution of contacts (<4.0Å closest atom–atom distance) between DAT and sydnocarb (indicated by orange-shaded areas) with the binding frequency summarized by the horizontal blue bars on the right panel. (**C**,**G**) Sydnocarb-binding poses captured in simulations with a snapshot taken every 4 ns. The ligand conformations are shown in cyan sticks. (**D**,**H**) MD-resolved final poses of sydnocarb (light orange vDW) observed at the end of MD Run1 and Run2. Results for MD simulation Run3 and Run4 can be found in Appendix A.

**Figure 4 biomedicines-09-00634-f004:**
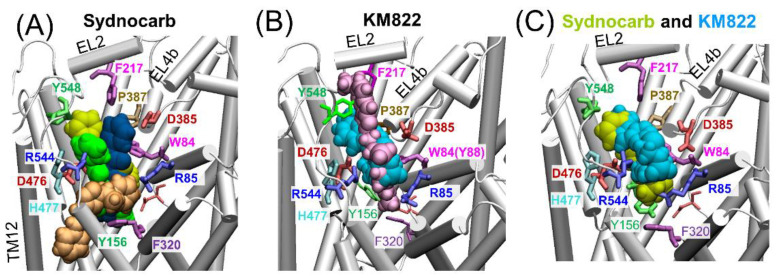
Comparison of MD-resolved ligand binding poses. (**A**) Sydnocarb-binding poses captured by four independent MD runs. Light orange, green, yellow, and dark blue vDW balls display representative poses, stabilized in MD simulations Run1, Run2, Run3, and Run4. Detailed results are shown in Figure 3 (Run1 and Run2) and Appendix A (Run3 and Run4). Note that three MD runs (Run2, Run3, and Run4) converged on a similar site. The binding pose resolved by Run1 differed significantly. (**B**) KM822-binding poses captured by two independent MD runs. Cyan and mauve vDW balls showed representative KM822-binding poses, stabilized in MD simulations KM822-Run1 and KM822-Run2, respectively. Detailed simulation results are shown in Appendix A. (**C**) Resemblance of sydnocarb and KM822 binding. Yellow and cyan vDW balls showed a representative sydnocarb-binding pose stabilized in MD simulation Run3 (Appendix A–D) and KM822-binding pose captured in KM822-Run1 (Appendix A–C). Representative binding poses were taken at the end of the MD simulations.

**Figure 5 biomedicines-09-00634-f005:**
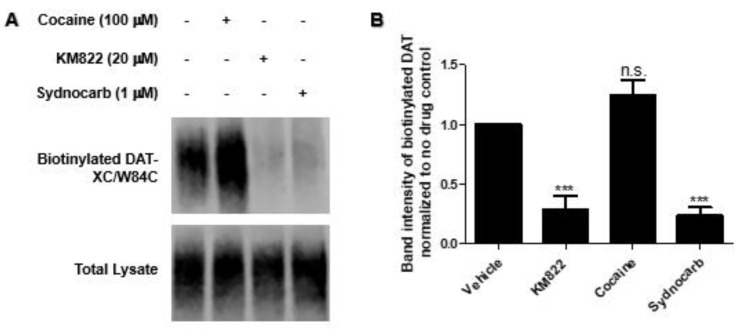
Analysis of the substituted cysteine scanning mutagenesis results. (**A**). Representative immunoblots of biotinylated DAT-XC/W84C mutant and total DAT protein in the presence of the vehicle, cocaine (100 µM), KM822 (20 µM), and sydnocarb (1 µM). (**B**). Quantification of the biotinylation data. The biotinylated DAT is normalized to the total DAT protein and then, the drug-treated biotinylated DAT band intensity is normalized to the untreated band intensity. The data represents four independent experiments. Statistical analysis was performed using one-way ANOVA with Dunnett’s multiple comparison to determine the significance between drug-treated and untreated samples. ***, *p* < 0.001; n.s., no significance.

**Figure 6 biomedicines-09-00634-f006:**
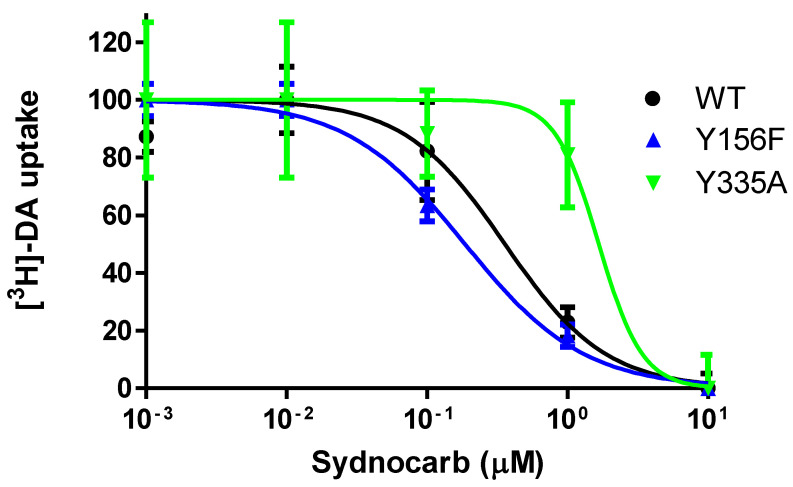
Dose-response assay of sydnocarb against DAT mutants Y156F and Y335A versus WT-DAT. Nonlinear regression analysis of the normalized response gave sydnocarb IC_50_ as 0.439 ± 0.135, 0.262 ± 0.118, and 1.419 ± 0.201 µM in WT-DAT, Y156F-DAT, and Y335A-DAT-transfected COS-7 cells, respectively. Averages and SEM were calculated from three independent experiments.

**Figure 7 biomedicines-09-00634-f007:**
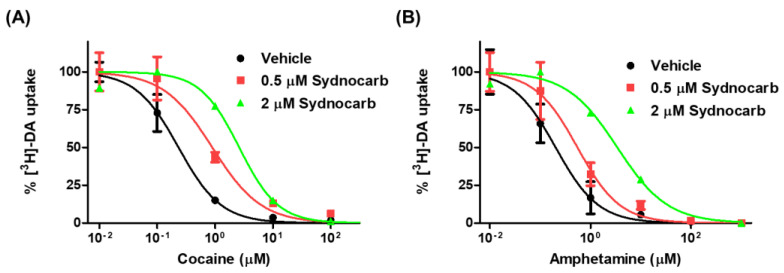
(**A**) Dopamine transport inhibition assay of cocaine in the absence and presence of 0.5 µM and 2.0 µM sydnocarb in hDAT-transfected COS-7 cells. IC_50_ values of cocaine are 0.177 ± 0.034 µM for the vehicle, 1.38 ± 0.24 µM in the presence of 0.5 µM sydnocarb (*), and 8.40 ± 0.99 µM in the presence of 2.0 µM sydnocarb (***). (**B**) Dopamine transport inhibition assay of amphetamine in the absence and presence of 0.5 µM and 2.0 µM sydnocarb in hDAT-transfected COS-7 cells. IC_50_ values of amphetamine are 0.166 ± 0.020 µM for the vehicle, 0.642 ± 0.094 µM in the presence of 0.5 µM sydnocarb (*), and 2.60 ± 1.3 µM in the presence of 2.0 µM sydnocarb (***). The figures were plotted using the average of three independent experiments, and IC_50_ mean and SEM was calculated using the same experiments. Results were normalized to the percent of the highest response in each group. One-way ANOVA with Dunnett’s post-hoc test was performed to determine the significance: * *p* < 0.05 and *** *p* < 0.001 when compared to vehicle.

## Data Availability

The data presented in this study are available on request from the corresponding author.

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
