# Peer review of "Functional Characterization of the Dopaminergic Psychostimulant Sydnocarb as an Allosteric Modulator of the Human Dopamine Transporter"

_biomedicines, 2021, doi:10.3390/biomedicines9060634_

Round 1

Reviewer 1 Report

see attached

Author Response

We appreciate the thoughtful critiques and have attached file with our response

Reviewer 2 Report

Excellent, important and well-executed study.

Author Response

We appreciate the positive reviews.

Reviewer 3 Report

Review for Biomedicines-1194997

This article characterized the "atypical" selective dopamine transporter (DAT) inhibitor Sydnocarb as an allosteric modulator using molecular simulations and dopamine uptake bioassays. This study is a natural extension of the author's previous work on allosteric modulator KM822(reference 17, 18, and 20).

The authors repeated the experiments on Sydnocarb and compared the two modulators on their binding sites in hDAT in silico. The simulation was carried out similarly as reported in reference 27 but resulted in two sites for each molecule.

Major concerns:

  1. The article did not demonstrate the resemblance between Sites I&II for Sydnocarb and Sites I&II for KM822. In other words, the chain of equivalence: Sydnocarb Sites II ~ KM822 Sites II ~ reported Site A2 was not established.
  2. Large difference of mean KM values in DA uptake kinetic assay (1.3 vs. 3.6 µM) were reported, yet no statistical difference found with only four repeats. Also, only one dose of Sydnocarb (1 µM) was tested in the assay vs. two doses of KM822 (1 and 5 µM) in reference 17.
  3. Lack of dose-response in bioassays. Only one concentration of Sydnocarb was applied in DAT-XC/W84C mutant SCAM analysis (1 µM) and DA transport inhibition assay for cocaine (0.5 µM) or amphetamine (0.5 µM).

Minor concerns:

  1. Figure 1A, non-fitting data for hSERT. The authors might want to discuss the highly varying inhibition results and thus poor-fitting in nonlinear regression for hSERT inhibition.
  2. Line 264 needs reference or discussion for Sydnocarb binding to the intracellular vestibule. Is it possible that Sydnocarb was internalized into neurons like nicotine and acted from the inside out?
  3. Line 289, MD simulations for the KM822 resulted in stabilizing at Site I, then transitioned to Site II within the 150 ns timeframe. Further discussion is needed for the absence of Sydnocarb transition or migration between Site I and Site II in hDAT.
  4. Figure S2, only one time-evolution result was plotted for KM822. However, the Method stated that at least two independent runs of 150 ns were performed. Were the results from 2 independent runs were the same for KM822?
  5. Figure 3, two binding poses of Sydnocarb in hDAT were reported from just two independent MD runs. Could more runs reveal even more binding poses?
  6. Figure 3, missing data for the last ~15 ns RMSD for Site II in panel E.
  7. Figure 3, including summarized binding frequencies (like in Figure S2), will help understand the contact between hDAT and Sydnocarb better.
  8. Line 356-358, W84/R85 was not the contributing binding residue with the highest probability for Sydnocarb or KM822. Many other high probability binding AA residues did not bind to cocaine. Why was W84 chosen for the SCAM analysis?
  9. Unit µ typos in lines 403 and 458.

Author Response

We appreciate the thoughtful critiques and have attached file with our responses.

Round 2

Reviewer 3 Report

In the revised manuscript, the authors have clarified most of the concerns listed in the last review.  The new version has added Figure 4 to link three binding sites.  The authors explained the limitations on the dose-response experiments and the selection of W84/R85 in the SCAM experiment (including alternative explanations).  The revision has significantly improved the quality and readability of the paper.

However, the author should have included more dose-response in the bioassays. It will further strengthen the conclusion, as the authors stated in the cover letter, but also helps to differentiate Synocarb and KM822 as atypical DAT blockers kinetically.  It will significantly increase the impact of the paper if the editors give an extension to add more dose-response to the results. 

Author Response

We have now added additional doses of sydnocarb as suggested. We show that sydnocarb dose-dependently affect all three assays - uptake kinetics by lowering the Vmax, cocaine inhibition by decreasing its potency, and similarly for amphetamine inhibition decreasing its potency. We agree the manuscript has been improved and strengthened by this addition. We thank reviewer and editor for giving us the time to include these important additional experiments.